# Novel Approach to the Treatment of Gypseous Soil-Induced Ettringite Using Blends of Non-Calcium-Based Stabilizer, Ground Granulated Blast-Furnace Slag, and Metakaolin

**DOI:** 10.3390/ma14185198

**Published:** 2021-09-10

**Authors:** Khaled Ibrahim Azarroug Ehwailat, Mohd Ashraf Mohamad Ismail, Ali Muftah Abdussalam Ezreig

**Affiliations:** School of Civil Engineering, Engineering Campus, Universiti Sains Malaysia, Nibong Tebal 14300, Seberang Prai Selatan, Pulau Pinang, Malaysia; Khaledazrog@student.usm.my (K.I.A.E.); Aliezreig@student.usm.my (A.M.A.E.)

**Keywords:** sulphate-bearing soil, soil stabilization, ettringite, gypseous soil, calcium-based stabilizer, non-calcium-based stabilizer

## Abstract

Gypseous soil is one type of expansive soil that contains a sufficient amount of sulphate. Cement and lime are the most common methods of stabilizing expansive soil, but the problem is that lime-treated gypseous soil normally fails in terms of durability due to the formation of ettringite, a highly deleterious compound. Moisture ingress causes a significant swelling of ettringite crystals, thereby causing considerable damage to structures and pavements. This study investigated the suitability of various materials (nano–Mg oxide (M), metakaolin (MK), and ground granulated blast-furnace slag (GGBS)) for the stabilization of gypseous soil. The results showed soil samples treated with 20% M-MK, M-GGBS, and M-GGBS-MK to exhibit lower swelling rates (<0.01% change in volume) compared to those treated with 10% and 20% of lime after 90 days of curing. However, soil samples stabilized with 10% and 20% binder of [(M-MK), (M-GGBS), and (M-GGBS-MK)] exhibited higher strengths after 90 days of soaking (ranging from 0.96–12.8 MPa) compared to those stabilized with 10% and 20% lime. From the morphology studies, the SEM and EDX analysis evidenced no formation of ettringite in the samples stabilized with M-MK-, M-GGBS-, and M-GGBS-MK. These results demonstrate the suitability of M-MK, M-GGBS, and M-GGBS-MK as effective agents for the stabilization of gypseous soil.

## 1. Introduction

Most of the geotechnical problems are believed to be caused by weak soils with low strength and high compressibility; such soils must be treated with suitable stabilizers/solidifiers to enhance their load-bearing capacity [1,2]. This deterioration in strength is clearly observed in the sulphate-bearing soil. Sulphate is a soil anion that forms from the dissolution of certain minerals like sodium sulphate (Na_2_SO_4_), gypsum (CaSO_4_∙2H_2_O), magnesium sulphate (MgSO_4_), anhydrite (CaSO_4_), and barite (BaSO_4_). Gypsum remains the major form of sulphate found in soil; however, sulphate ions are also sourced from the oxidation of certain soil minerals, such as pyrite (FeS_2_) [3,4]. Sulphate-bearing soil, also known as gypseous soil, has a high capacity for swelling and a low strength when wet. This soil causes serious damage to structures, such as roads, airports, single-storey buildings, and garden walls [5,6,7].

Soil stabilization is the alteration of the undesired soil characteristics through certain chemical and physical processes to meet specific engineering purposes [8]. Despite the availability of numerous methods of soil stabilization, chemical stabilization with additives remains the common way of preventing soil swelling with additives, with lime and cement being the most utilized chemical additives for soil stabilization [1,3,9].

The treatment of sulphate-rich soil with lime or cement has been reported to cause volume expansion [3,5,7,10,11,12]. The three main stages of lime-soil reaction are cation exchange, flocculation, and pozzolanic reaction [7,13]. Furthermore, it has been demonstrated that soil contains clay (>10 wt dry soil weight) and >1% sulphate content after stabilization with cement or lime [4]. Clay releases ammonia at pH values > 10.5 and this ammonia interacts with Ca^2+^ (contained in lime or cement), soil sulphate ions, and available soil water to form calcium silicate trisulphate hydrate (ettringite) [11,14,15,16]. This ettringite can be converted to calcium-silicate-hydroxide-sulphate-carbonate-hydrated material (thaumasite) at temperatures < 15 °C [17,18,19]. The reaction of sulphate-rich soil with calcium-based soil stabilizers is explained in Figure 1.

Ettringite forms fibrous crystals and damages the soil structure through mineral expansion during its precipitation. These sulphate minerals expand significantly when subjected to the hydration process [19,20]. Both hydration reactions and crystal growth causes significant heaving in sulphate-rich soils [21,22]. Sulphate-containing clay soils cause swelling with the application of calcium-based soil stabilizers. Furthermore, the destruction of calcium aluminosilicate hydrates (CSH) causes a reduction in long-term strength [23].

Civil engineers have recently focused on the development of building materials with low cement contents as a way of reducing greenhouse emissions, climate changes, and carbon footprint of the industry. The processes involved in the production of lime and cement contribute significantly to environmental pollution. For instance, the production of 1 ton of cement leads to the emission of almost 0.95 tons of CO_2_ and requires almost 5000 MJ of energy to complete [2,24,25,26]; for lime production, these values are ~0.79 tons of CO_2_ and ~3200 MJ of energy [27]. This demands the consideration of waste materials as a complete or partial replacement for conventional binders, as proposed by various researchers. GGBS is a waste material from the iron industry and has been considered a suitable material for soil stabilization. In terms of CO_2_ emissions and energy consumption, producing 1 ton of GGBS requires only 1300 MJ of energy and emits only 0.07 ton of CO_2_ [28].

Novel methods should be able to repress heaving and ettringite formation in sulphate-rich soils. Materials that exhibit pozzolanic properties, such as GGBS and MK are believed to be suitable sulphate soil stabilizers, because such materials generally consume lime, thereby reducing its availability for the formation of expansive products while enhancing the strength of the soil.

Metakaolin is a highly pozzolanic, reactive, and supplementary cementitious material that adapts to ASTM C 618 [29] and AASHTO M295 specification [30]. It is unique because it is neither an industrial by-product nor completely natural. Metakaolin is manufactured from natural minerals specifically for cementing purposes; it is manufactured from kaolin clay that has been calcined under high temperature (600–800 °C) to create an amorphous aluminosilicate that is reactive in concrete [31,32,33]. Sulphate ions intrusion into concrete weakens the cement paste-aggregate bond, thereby causing severe damages, such as expansion and extensive cracking. An increase in MK content (5–20%) has been reported to reduce mortar expansion when used with intermediate and high C_3_A content cement [33].

GGBS can serve as partial or complete replacement for cement or lime to prevent or reduce ettringite formation in stabilized sulphate soils. The alumina and silica contents of GGBS can quickly react with the calcium content in soil to form a cementitious gel, thereby reducing or preventing ettringite formation [8,28,34]. GGBS can also reduce water availability and permeability through the formation of a denser cementitious matrix, which improves resistance to both external and internal sulphate attacks [28,35,36].

Reactive magnesia (M) is a more effective substance in GGBS activation compared to lime, as it facilitates a higher rate of strength development [6,8,37,38,39,40]. Reactive M is commonly obtained from the calcination of magnesite at temperature range of 700–800 °C and high capacity in absorbing carbon dioxide (CO_2_) [2,41]. Reactive M is used at low concentration during GGBS activation, hence the overall CO_2_ emission during the reaction of M with GGBS is considered relatively lower than that of lime and cement. For example, the production of 1 ton of M-GGBS with M at the optimum M:GGBS ratio of 1:9-1:4 only emits about 0.20–0.34 tons of CO_2_ [27,28,37]. The study by Li et al. [6] also showed that M-GGBS binder protected gypseous soil samples from swelling and contributed to better strength after soaking compared to Portland cement (PC).

The main purpose of this research is the establishment of a new method to stabilize soils subjected to internal sulphate attack by the gypsum (CaSO_4_) by using combined of nano-MgO, MK, and GGBS. For this purpose, different tests (flexural strength (FS), linear expansion (LP) tests, and unconfined compressive strength (UCS)) were performed before and after exposure to water. In this study, two steps have been taken to achieve the objectives. Step 1: running compaction characteristics for each mixture, and step 2: performing UCS, FS, and LP tests before and after sulphate exposure. The samples were cured for 7, 28, and 90 days.

## 2. Materials and Methods

### 2.1. Materials

The materials utilized for this research were kaolin clay (K), hydrated lime (L), calcium sulphate (gypsum) (G), (M), (MK), and (GGBS). Kaolin clay was supplied by KAOLIN (Malaysia) SDN BHD under the brand name MK40 as a white, finely ground, odorless powder. Kaolin clay was used because (i) it is a major component of minerals found in natural clay; (ii) it exhibits a uniform and consistent mineralogy; (iii) it has a low cation exchange capacity [5,40]; (iv) it has a higher alumina content than most other costly minerals and can thus release more alumina when the pH is high, thereby participating in the formation of ettringite which increases susceptibility to sulphate attack [10,28]. For these reasons, kaolin clay is a suitable control soil in the soil stabilization process. Table 1 summarizes the major properties of kaolin clay. A hydrometer test of clay was conducted according to BS EN ISO 17892–4:2016 [42] with the grading curve shown in Figure 2.

L, G, and MK were collected from Sungai Jawi, 14200, Penang, Malaysia. However, G (CaSO_4_∙2H_2_O) was selected in this research due to its much lower solubility compared to other sulphate types (potassium sulphate, magnesium sulphate, and sodium sulphate), as presented in Table 2. Moreover, it is one of the sulphates that contain calcium, and it is logical that it would form ettringite if a non-calcium-based stabilizer had been used, as presented in Table 2.

Metakaolin is produced from kaolin, a natural material. It is a pozzolanic material that results from the heating of kaolin at 700–750 °C; its classification as a pozzolanic material follows the ASTM C 618 standard. MK exhibits a high surface area and amorphous structure which contributes to its high pozzolanic reactivity [33,43]. The water content of kaolin is reduced during the heating process, thereby altering its structure and forming metakaolin, an amorphous aluminosilicate (Al_2_O_3_∙2SiO_2_) as seen in Equation (1) [44].
(1)$$Al_{2}O_{3}\cdot 2SiO_{2}\cdot 2H_{2}O\overset{700-750\left(\mathrm{heat}\right)}{\to }Al_{2}O_{3}\cdot 2SiO_{2}$$

GGBS is a waste-product from the pig-iron manufacturing process and is formed through the rapid cooling of molten iron slag to retain its amorphous structure, followed by grinding to increase its specific surface area. The GGBS used in this study was collected from MDC Sungai Pentani Company, Malaysia. It was chosen to reduce the rate of ettringite formation through the provision of more Al and Si, which react with Ca^2+^ to form complex cementing gels [10,35,36]. The denser structure and lower calcium ion content of GGBS should contribute to superior sulphate resistance [8,28,45]. A hydrometer test of MK and GGBS with the grading curve is shown in Figure 2.

In this research, M was chosen, as it is a green and low-carbon clay soil stabilizer. It was collected from Hang Zhou Jiu Peng New Material Co., Ltd., Hangzhou, China as a white, fine, crystalline powder. The production of reactive M requires low temperatures, consumes less fuel, and emits less CO_2_ than that of Portland cement (PC) [6,28]. Nanostructured binders have been receiving more attention recently because the main hydrate cement is also a natural nanostructured material [24,46]. Furthermore, it reduces sulphate-induced expansion and has higher reactivity and a lower crystallinity [6,34], thereby increasing the rate of reaction with water [38,47].

The chemical content of the materials was determined using X-ray fluorescence test (XRF) as listed in Table 3.

### 2.2. Samples Preparation

Sulphate-bearing soil or gypseous soil was prepared artificially by mixing kaolin clay with 10% gypsum (by dry weight of soil). The concentration of sulphate was determined as the worst case according to AASHTO [22,48,49]. The risk of different sulphate levels is summarized in Table 4.

Cylinder samples of 50 × 100 mm (diameter × height) dimension were prepared as reported by [5,6,50] for tests of linear expansion (LP), flexural strength (FS), and unconfined compressive strength (UCS). Each mixture system was compacted with an optimum moisture content (OMC) and maximum dry density (MMD) following the BSEN 13286–2:2010 standard [51], as shown in Table 5. After compaction of samples, each sample was covered using cling film to reduce moisture loss.

The total binder contents were fixed at 10% and 20% based on the weight of the soil (see Table 6) for each system (unary, binary, and ternary). This was achieved using activator (L) calcium-based stabilizer and (M) non-calcium-based stabilizer at dosages of 10% and 20%, with (G) calcium sulphate dosed at 10% (as a worst case) into kaolin clay. The ratios of M stabilized with (MK and GGBS) (Figure 2 shows a comparison of particle size distribution between MK and GGBS) were set as 1:3, 1:1, and 3:1 in a binary system, and 1:05:05, 1:1:2, 1:2:1, and 1:2.5:0.5 in a ternary system. Figure 3 summarizes the experimental process. In total, 462 cylindrical samples were prepared: 207 for testing UCS, 138 for LP, and 117 for FS.

### 2.3. Experiments

#### 2.3.1. Linear Expansion Test (Swelling) (LP) Test

Two cylindrical samples were prepared for each mix proportion, and were subjected to curing for 7, 28, and 90 days to establish their vertical swelling ratio (%). Swelling readings were recorded every 24 h until no significant swelling ratio was observed after partial soaking in water. The LP test was conducted following the BS EN 13286–49:2004 standard [52].

#### 2.3.2. Unconfined Compressive Strength (UCS) Test

The UCS test was performed according to BS EN ISO 17892–7:2018 [53] for each mix proportion; three cylindrical samples were tested for compressive strength before and after soaking. “Before soaking” implies that the samples had been cured for 7, 28, and 90 days without having been immersed in water, while “after soaking” implies after the linear expansion test, which was 52 days after the preparation of the samples for all soil stabilization systems. The samples were subjected to a constant compression strain rate of 1 mm/min until failure.

#### 2.3.3. Flexural Strength (FS) Test

For this test, specimens were prepared as for the UCS and LP tests. A three-point bending test was used to determine the FS for kaolin clay and the optimum mixing of sulphate-bearing soil mixed with the calcium-based and non-calcium-based stabilizers. For each test, three samples were prepared using a compaction mould at the optimum moisture content (OMC) and maximum dry density (MMD); the curing process was done for 7, 28, and 90 days. The ASTM D1635 was followed during the flexural strength test; the test was performed at a constant strain rate of 0.1 mm/min. Equation (2) was used to calculate the flexural stress for the outer layer of each cylindrical specimen.
(2)$$FS=PL/\pi r^{3}$$
where FS = flexural strength; P = maximum load applied; r = sample radius; L = support span.

#### 2.3.4. Microstructure Analysis (SEM/EDX)

The samples were subjected to surface morphology analysis using scanning electron microscopy (SEM); the elemental content of the samples was also elucidated using energy dispersive spectroscopy (EDS). Small portions of soil specimens were collected from carefully hand-broken samples after UCS and then dried for 24 h at 105 °C prior to testing.

## 3. Results and Discussion

### 3.1. Unconfined Compressive Strength (UCS) Test

#### 3.1.1. Effects of M-MK on UCS

Figure 4 and Figure 5 depict the effects of L and M additions on the UCS values after 7, 28, and 90 days of curing for the gypseous soil samples. The UCS values for M-treated samples increased from 885.33 to 1108.67 kPa when the M content was increased from 10% to 20% without the presence of sulphate at 90 days of curing. Samples treated with L and M (10% and 20%) showed a decrease in UCS values in the presence of sulphate (see Figure 4 and Figure 5). Furthermore, after soaking, the UCS values deteriorated from 885.33 to 635 kPa for M10%, (reducing by 28.27%) and 1108.67 to 1035.33 kPa with 20% M content in the presence of sulphate (reducing by 6.62%). However, this deterioration, as compared to that of the lime stabilizer, was much less. The observed deterioration in the UCS results can be attributed to ettringite production, the growth of which between the particles of the sample soil would lead to the destruction of the soil structure. In addition, the residual sulphate chelates formed calcium in CSH through a decalcification process, which accounted for the lower strengths of the soaked and unsoaked samples. However, the higher strength of the M-treated samples might have been due to the hydration process and formation of pozzolanic products that might have improved the strength of the bonds between the soil particles.

Figure 4 and Figure 5 show the effects of the MK content on enhancing the USC values for the M-treated samples before soaked after 7, 28, and 90 days of curing in the presence of sulphate. For example, Figure 5 indicates that the sample’s UCS values of the samples treated with 7.5M-12.5MK at the age of 90 days increased from 756.33 kPa to 1172.67 kPa as compared to samples at the age of 7 days, where UCS increased by 55.05%. This impact was also observed for 5M-15MK (increased from 591 to 858.33 kPa), which was improved by 45.23%, and 5M-5MK (increased from 736 to 973 kPa), which was enhanced by 32.2%. These improvements could be due to pozzolanic product (CSH) formation, which would have improved the USC value for the samples with higher M-MK content. However, all values of UCS decreased after soaking, except soil samples treated with binder 10% (5M-5MK), increased from 973 kPa to 1145 kPa at the age of 90 days, where improved by 17.68%. The enhancement of the hydration process could explain this result after immersing in water. A possible reason for the loss in strength of the M-MK stabilized soil after soaking in water, such as 7.5M-12.5MK, is the increase of metakaolin content, leading to reduce in hydrotalcite formed. This phenomenon led to cracks during water immersion, resulting in a compromised microstructure of the M-MK stabilized soil and becoming relatively brittle. These results agree with the report by [54], where hydrotalcite formation positively contributed to the greater mechanical property of the binder.

#### 3.1.2. Effects of M-GGBS on UCS

For the M-GGBS stabilizer, all specimens showed enhancement of their UCS values. Figure 6 and Figure 7 presented the maximum UCS of the M-GGBS-stabilized soils at the M:GGBS ratio of 1:3 when a 20% binder was used to stabilize the sulphate-bearing soil: the UCS values were 3327.33, 7125, and 12,115 kPa after curing for 7, 28, and 90 days, respectively. This improvement may have been because of the high content of highly reactive M, and because an excessive residual can negatively impact the strength of stabilized soils. Nevertheless, the UCS values of the stabilized samples before and after soaking indicated enhancement in the mechanical performance of the samples after wetting, except ratios of 3:1 and 1:1 when used to treat gypseous soil samples with 10% M-GGBS-stabilizer (see Figure 6); for example, the samples treated with 5M-5GGBS deteriorated by 13.02%, 33.11% and 29.76% at 7, 28 and 90 days of curing, respectively. The UCS of the treated soil sample with 2.5M:17.5GGBS was 11,275 kPa after soaking, which was significantly higher than that before soaking (914.23%) after 7 days of curing. After soaking, the 5M-15GGBS ratio of 1:3 exhibited an optimum UCS value of 12,831 kPa, which was increased by 5.91% after 90 days of curing. However, the UCS values showed a decline at higher M-GGBS ratios. These findings agree with the report by [6], where highly reactive M was found to have better activation efficiency with GGBS compared to low-reactive M; therefore, lower M-GGBS ratios are recommended to achieve higher UCS values.

#### 3.1.3. Effects of M-GGBS-MK on UCS

Figure 8 and Figure 9 show that all ternary binder compositions improved the UCS values in the stabilized gypseous soil. The highest values of UCS obtained by the M-GGBS-MK stabilizer ratio of 5:12.5:0.5 were 1770.33, 3196, and 4007.33 kPa at the ages of 7, 28, and 90 days, respectively (see Figure 9). However, the lowest enhancements after 7, 28, and 90 days of curing were 581, 734, and 994 kPa, respectively, when a stabilizer–binder ratio of 5:2.5:2.5 was used (see Figure 8). The UCS decreased with the reduction in the GGBS ratio.

After soaking, the M-GGBS-MK ratio of 5:2.5:2.5 improved the UCS values among all the M-GGBS-MK-stabilized soils, an increase of 52.49%, 25.47%, and 5.63% at the ages of 7, 28, and 90 days (see Figure 8). This enhancement in the UCS values might have been due to the hydration process following the prolonging of the curing time from 7 days to 90 days; this should ensure proper hydration, improved strength, and resistance against sulphate attacks [8]. Furthermore, the production of more cementitious gels, such as Calcium silicate hydrates (CSH), calcium aluminate hydrates (CAH), and calcium aluminosilicate hydrates (CSAH), contributed to the consumption of the available calcium, leading to the inhibition of ettringite formation [5].

### 3.2. Linear Expansion Test (Swelling) (LP) Test

Figure 10, Figure 11, Figure 12, Figure 13, Figure 14 and Figure 15 illustrate the typical swelling plots for K-M clay and K-L-G and K-M-G clay systems dosed with 10% wt of gypsum and stabilizer with 10% and 20% wt of L and M for observatory periods of 7, 28, and 90 days, respectively. Swelling was observed immediately after soaking the cylinder samples in water, which was sustained throughout monitoring until the cessation of swelling. A higher rate of swelling was observed compared to the reported expansion in lime-stabilized sulphate-bearing soils [5,12,34].

Figure 10 and Figure 13 show the highest degree of swelling obtained with a calcium-based stabilizer (10% wt): Which was 25.226% with 10% lime after 7 days curing; meanwhile, the lowest swelling was observed with 20% lime: 18.313% after seven days of curing. Nevertheless, the comparison of the swelling values of the treated soils between the calcium-based stabilizer and non-calcium-based stabilizer clearly showed that the non-calcium-based stabilizer (M) had a significant effect on restricting the swelling values. For example, when 10% and 20% M were used, the swelling values were 0.18% and 0.466%; hence, the swelling was restricted to 99.28% and 97.45%. This result was due to the absence of calcium in non-calcium-based stabilizer (M), which led to the suppression of ettringite formation.

#### 3.2.1. Effects of M-MK on Swelling

Utilizing the M-MK stabilizer resulted in a restriction in the volume change (swelling) for all gypseous soil specimens, as presented in Figure 10, Figure 11, Figure 12, Figure 13, Figure 14 and Figure 15. The lowest minimum values of swelling, achieved with a 5M-5MK ratio of 1:1, were 1.68%, 0.401%, and 0.197% after 7, 28, and 90 days of curing, respectively, with the swelling inhibited 93.34%, 97.82%, and 93.61%, respectively, as compared to 10% K-L-G. Furthermore, the 20% M-MK-stabilizer led to a low volumetric change after 90 days of curing at 0.025%, 0.004%, and 0.059%, with the swelling suppressed to 98.87%, 99.82%, and 97.34%, respectively, as compared to 20% K-L-G. This reduction in swelling can be attributed to the restriction of ettringite formation and the production of more CSH compounds in the absence of calcium.

#### 3.2.2. Effects of M-GGBS on Swelling

Figure 10, Figure 11, Figure 12, Figure 13, Figure 14 and Figure 15 present the vertical volume change (swelling) of gypseous soil treated with the M-GGBS stabilizer. The maximum vertical volume change was 7.293%, obtained after applying a 7.5M-2.5GGBS ratio of 3:1 over 7 days of curing. However, all other specimens exhibited low swelling. For example, the swelling of soil samples stabilized with 5M-15GGBS and 2.5M-17.5GGBS was 0.001% and 0.005% after 90 days of curing, with swelling magnitude almost wholly suppressed to 99.95% and 99.77%. The reduction in the swelling values was because of the use of GGBS, which exhibited superior sulphate resistance along with its denser structure and lower Ca^2+^ content [8,45].

#### 3.2.3. Effects of M-GGBS-MK on Swelling

Figure 16, Figure 17 and Figure 18 reveal that the use of the M-GGBS-MK stabilizer (10 and 20 wt. %) modified the behavior of the volumetric change and significantly reduced the extent of swelling of the gypseous soil. After 28 days of observation, the swelling magnitudes for 5M-2.5GGBS-2.5MK, 5M-5GGBS-10MK, 5M-10GGBS-5MK, and 5M-12.5GGBS-2.5MK were 0.105%, 0.254%, 0.208%, and 0.04%, respectively, with swelling values suppressed to 98.79%, 97.09%, 97.62%, and 99.54% respectively. Nevertheless, the long curing period (90 days) produced further declines in the swelling rate, thereby achieving minimum swellings of 0.005%, 0.067%, 0.043%, and 0.001%, where the swelling was roughly inhibited to 99.77%, 96.92%, 98.06%, and 99.95% respectively.

Results from the linear expansion test indicated that swelling behavior disappeared in the high-sulphate soil, which was treated with the M-GGBS-MK-stabilizer. The volume changes of the treated soils were less than those of the lime-treated soils, indicating that the swelling characteristics were enhanced in the treated high-sulphate soils.

These results indicate that the M-GGBS-MK-stabilized soil resisted sulphate attacks better than the soil stabilized with lime. Moreover, the results showed that the swelling of the soil stabilized with lime was better than that of soil stabilized with M-GGBS-MK at prolonged curing periods. The phenomenon can be explained by two factors: (1) A larger amount of calcium in the lime-stabilized soil contributed to ettringite formation in the presence of sulphate and (2) the consumption of calcium by sulphate, which led to a reduction in the formation of cementitious material such as CSH, CAH, and CSAH. Adeleke et al. [5] reported that there are various levels of swelling. In this study, the mixing of K-10L-10G and K-20L-10G soils, give the maximum expansion of 25.226% and 18.313%. Based on the level of swelling as shown in Table 7, the soil samples (K-10L-10G and K-20L-10G) can cause excessive swelling when they come into contact with water because the maximum expansion is more than 2%.

### 3.3. Flexural Strength Flexural (FS)

Flexural load curves for the kaolin clay and sulphate-bearing soil along with the calcium-based stabilizer (L) and non-calcium-based stabilizer (M) with the inclusion of MK, GGBS, and GGBS-MK ratios of 10% and 20% after curing for 7, 28 and 90 days, are shown in Figure 19 and Figure 20. The kaolin clay curve exhibited a flexural strength of 132.44 kPa after curing for 90 days. In contrast, the curves produced by the M stabilizer mixtures showed a slight improvement with and without the presence of sulphate, which is a direct measure of the flexural strength of the specimens. When compared to specimens treated with the L stabilizer, the flexural strength was less with the presence of sulphate. Furthermore, using 10% and 20% of the M-MK, M-GGBS, and M-GGBS-MK stabilizers significantly improved the peak flexural load; this was more significant for the 20% M-GGBS content. The flexural strength of the M-GGBS mixture with 10% and 20% content reached 2253.09 and 4627.53 kPa after curing for 90 days, respectively, gaining values 472.4% and 522.62% higher than those of the L stabilizer (393.6 and 743.23 kPa) using 10% and 20% mixtures, respectively, after 90 days of curing.

### 3.4. SEM and EDX Analysis

The SEM images of the gypseous soil–lime mixture after 90 days of curing showed that a significant amount of crystalline mineral needles (ettringite) had formed, as shown in Figure 21a and Figure 22a. This resulted in a decrease in strength, a high capacity for swelling, and the destruction of the soil specimen’s structure. The improvement in the gypseous soil samples was determined by measuring the Si/Al ratio (%) utilizing EDX to explain the strength development. Higher Si/Al ratios generally led to the formation of more Si-O-Si bonds which further increased the mechanical strength of the soil sample [55]. The Si/Al ratio in the (10% and 20%) L-G specimens obtained from the EDX pattern ranged from 1.06 to 1.12, which was low due to the ettringite formation and high porosity in the microstructure of the samples. The EDX pattern of the gypseous soil–lime mixture and its elemental composition is represented in Figure 21b and Figure 22b.

Figure 23a and Figure 24a show SEM observation images of the sulphate soil–M-G mixture, revealing spherical and irregular particles of magnesium silicate hydrate (MCH). The EDX information showed a slight increase in the Si/Al ratio (1.1 and 1.14) for M-G-stabilizer specimens with 10% and 20% compared to the L-G specimens (1.06 and 1.12), which was probably due to the absence of ettringite and the formation of MSH product. Figure 23b and Figure 24b, respectively, present the EDS spectrum and elemental composition of the gypseous soil–M-G mixture.

Figure 25a, Figure 26a and Figure 27a show the SEM observation images of M-GGBS, M-MK, and M-GGBS-MK, respectively. Regarding the surface features and particle shape, M-GGBS differed significantly from M-MK and M-GGBS-MK. M-MK and M-GGBS-MK particles were spherical, as shown in Figure 25a and Figure 27a. The surface of the particles was shiny and devoid of dust. For the M-GGBS particles, they were irregular and angular, and the presence of plate-shaped particles was evident (Figure 26a), which had rough, gritty surface textures. Figure 25a, Figure 26a and Figure 27a show the surface morphologies of the soil–M-GGBS, –M-MK, and –M-GGBS-MK specimens after 90 days of curing. As indicated, no ettringite formation was observed. This proves that the addition of M with GGBS and MK as a stabilizing agent contributed to the formation of CSH gel significantly. Some crystalline structures, such as MSH (magnesium silicate hydrate), were also observed. The pozzolanic activity occurring between M, GGBS, and MK as a stabilizer and clay was attributed to the formation of CSH gel. Based on the results derived from the EDX spectrum, the highest Si/Al ratios (1.22, 1.59, and 1.28) were observed in the 7.5M-12.5MK, 5M-15GGBS, and 5M-12.5GGBS-2.5MK samples, respectively. This improvement was due to GGBS and MK dissolving in an alkaline activator to generate extra soluble alumina and silica, which increased the Si/Al ratio of the specimen, leading to the complete activation of the particles producing calcium–alumino–silicate hydrate (CASH) in the matrix. Figure 25b, Figure 26b and Figure 27b show the EDS spectrum and elemental composition of the gypseous soil–M-GGBS, –M-MK, and –M-GGBS-MK mixtures, respectively.

The increase in UCS value with prolonged curing periods could have been due to cementing gel formation from the interaction of the soil with the nanoparticles, which improved the UCS of the treated samples as a result of the improved soil particle bonding [56,57].

## 4. Conclusions

Sulphate-bearing soils cause major issues in pavement and various civil engineering infrastructures due to their causing significant swelling and strength damage. The results of this study portrayed the impacts of (M) as a non-calcium-based stabilizer, metakaolin, and GGBS as abundant by-products in treating gypseous soil with a high level of sulphate. Based on the results of the study, the following conclusions were made:The minimum swelling values of M-MK-, M-GGBS-, and M-GGBS-MK-stabilized gypseous soils with 20% doses (0.004%, 0.001%, and 0.001%) were lower than that of lime-stabilized soil.The optimum strength values of M-MK-, M-GGBS, and M-GGBS-MK-stabilized soils after soaking with 20% (0.96, 12.8, and 3.72 MPa) were notably higher than that of lime-stabilized soil (0.9 MPa).The period of curing significantly affected the sample’s resistance to sulphate attack due to the impact of the hydration process. Extension of the curing time from 7 to 90 days caused adequate hydration and greater strength in the presence of sulphate (gypsum).SEM and EDX analysis showed no evidence of ettringite formation in the soil samples stabilized with M-MK-, M-GGBS-, and M-GGBS-MK.The formation of cementing gels such as MSH, MAH, CSH, and CAH improved the UCS values of the stabilized samples, as these gels occupied the voids and improved the bonding between the soil particles. As a result, the treated soils’ UCS values were improved.The M-MK, M-GGBS, and M-GGBS-MK were demonstrated as suitability and effective agents for the stabilization of gypseous soil.

## Figures and Tables

**Figure 1 materials-14-05198-f001:**
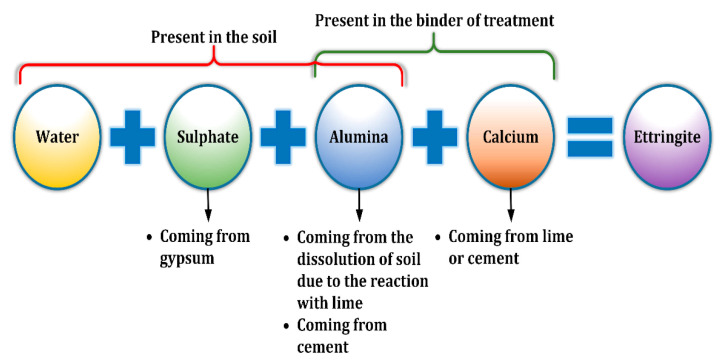
Schematic explaining the reaction of sulphate-bearing soil treated by cement and lime.

**Figure 2 materials-14-05198-f002:**
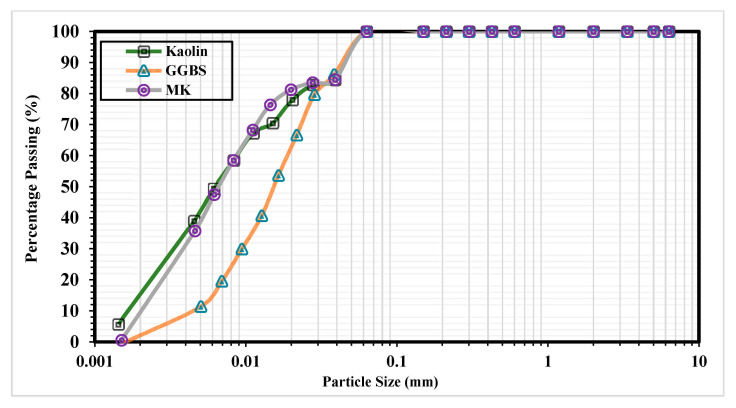
Particle size distribution (PSD) of the kaolin clay, MK, and GGBS used in this study.

**Figure 3 materials-14-05198-f003:**
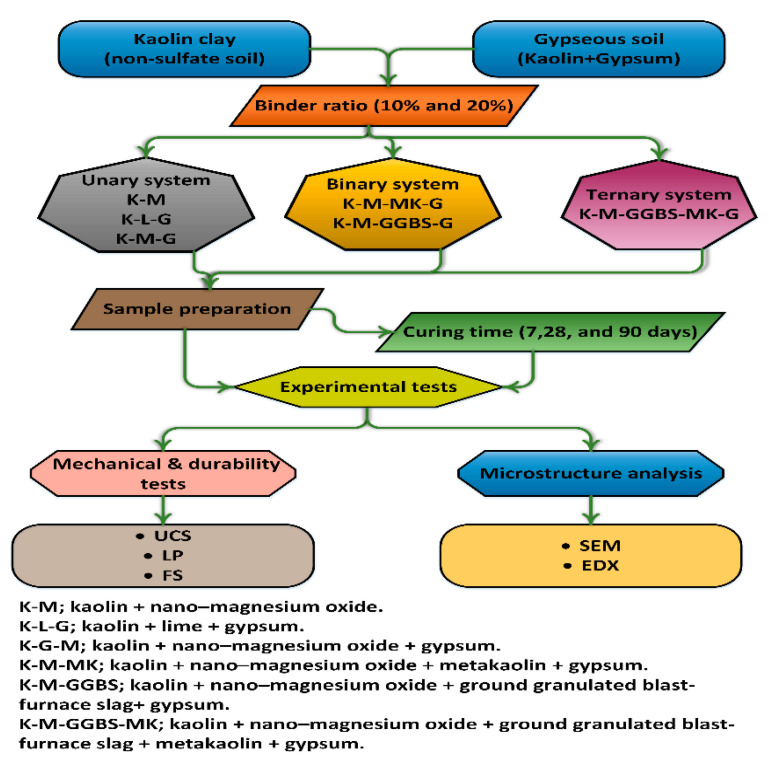
Schematic representation of the experimental procedure.

**Figure 4 materials-14-05198-f004:**
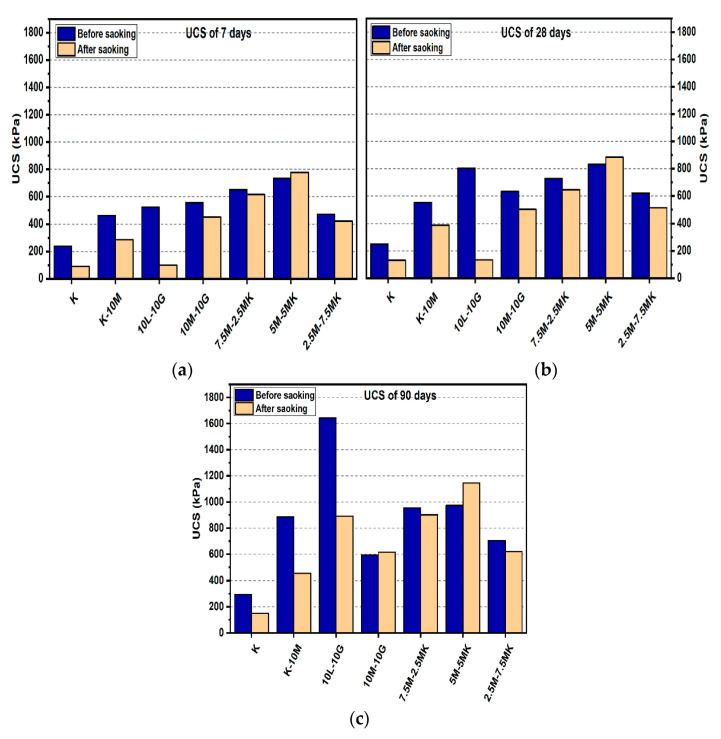
UCS of soils stabilized with 10% of L, M, and M-MK before and after soaking in the presence of sulphate: (**a**) after curing for 7 days, (**b**) after curing for 28 days, (**c**) after curing for 90 days.

**Figure 5 materials-14-05198-f005:**
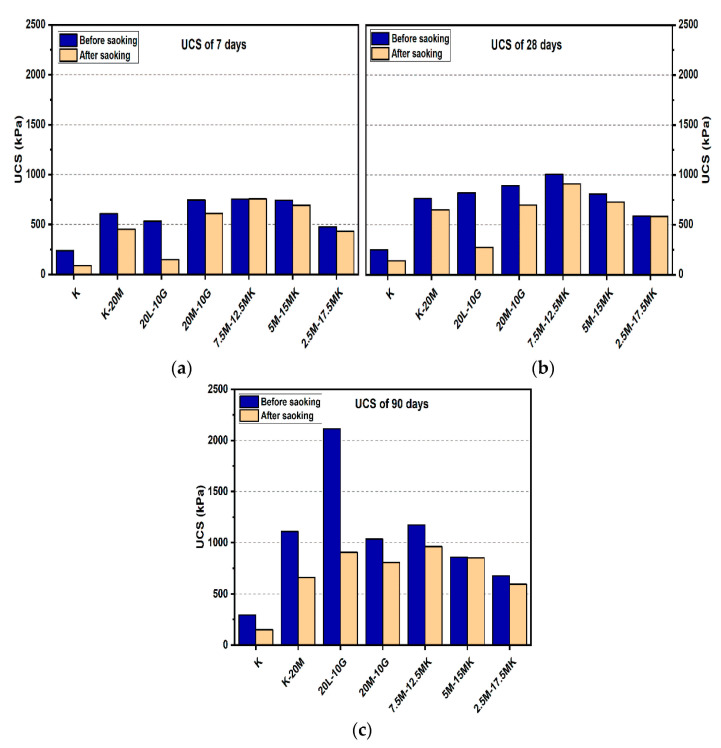
UCS of soils stabilized with 20% of L, M, and M-MK before and after soaking in the presence of sulphate: (**a**) after curing for 7 days, (**b**) after curing for 28 days, (**c**) after curing for 90 days.

**Figure 6 materials-14-05198-f006:**
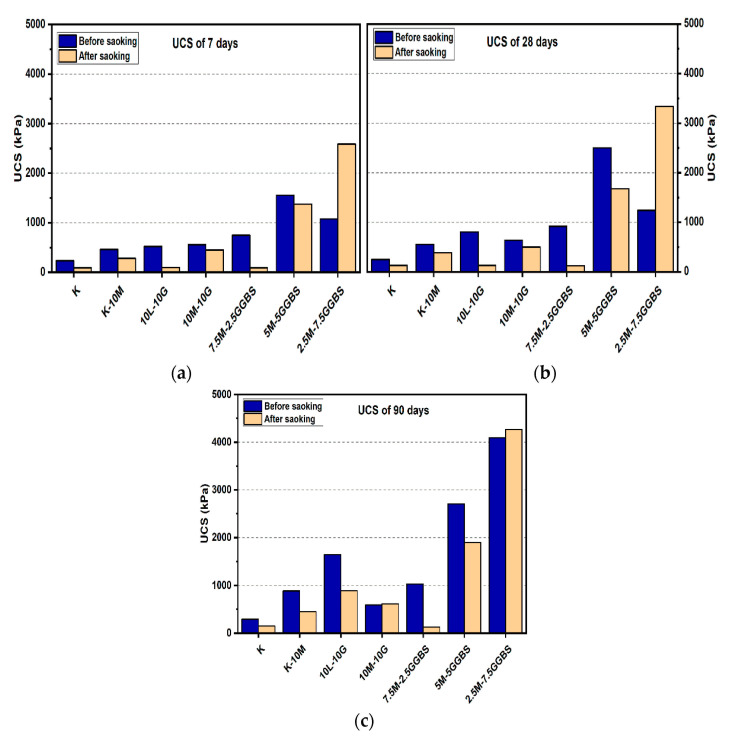
UCS of soils stabilized with 10% of L, M, and M-GGBS before and after soaking in the presence of sulphate: (**a**) after curing for 7 days, (**b**) after curing for 28 days, (**c**) after curing for 90 days.

**Figure 7 materials-14-05198-f007:**
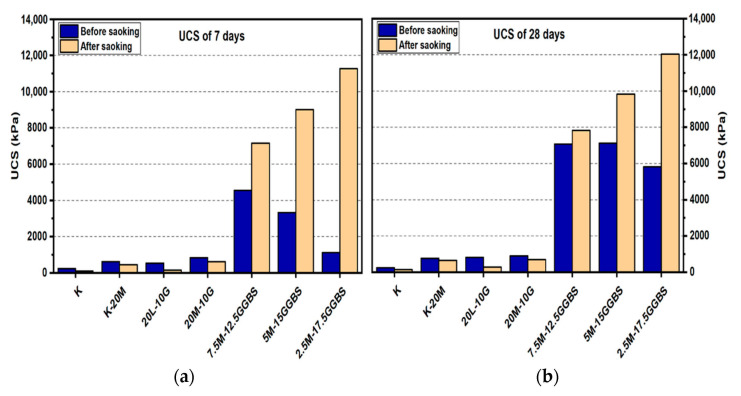
UCS of soils stabilized with 20% of L, M, and M-GGBS before and after soaking in the presence of sulphate: (**a**) after curing for 7 days, (**b**) after curing for 28 days, (**c**) after curing for 90 days.

**Figure 8 materials-14-05198-f008:**
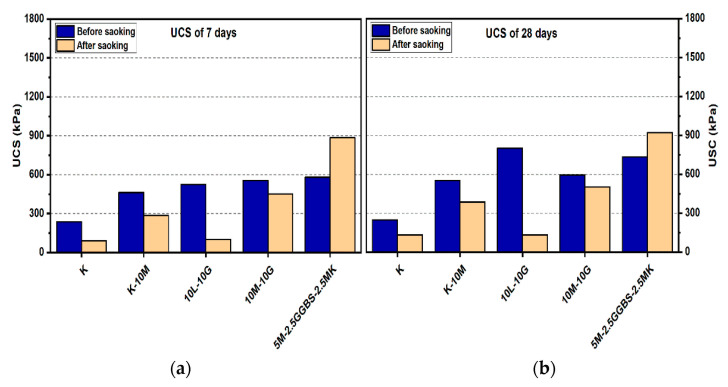
UCS of soils stabilized with 10% of L, M, and M-GGBS-MK before and after soaking in the presence of sulphate: (**a**) after curing for 7 days, (**b**) after curing for 28 days, (**c**) after curing for 90 days.

**Figure 9 materials-14-05198-f009:**
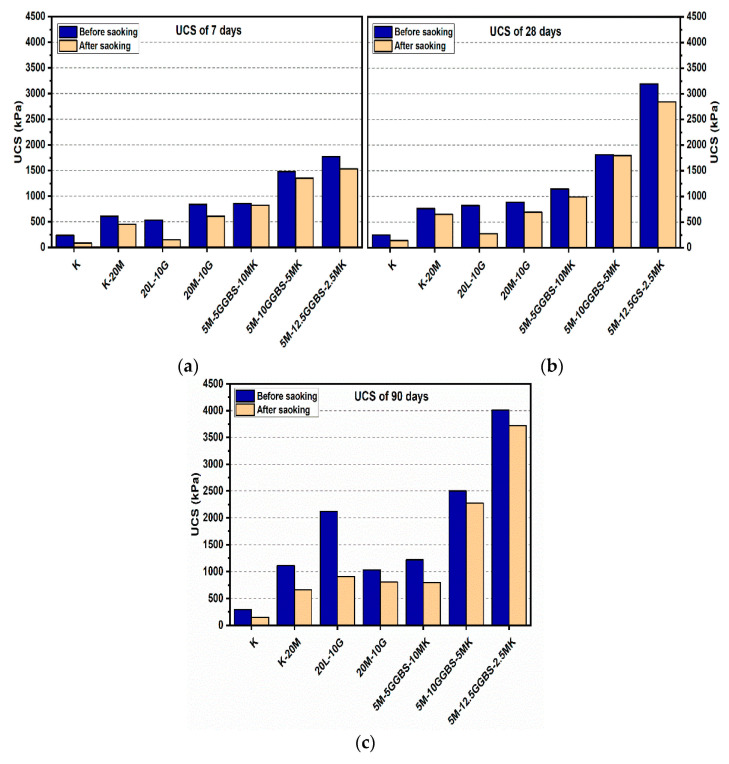
UCS of soils stabilized with 20% of L, M and M-GGBS-MK before and after soaking in the presence of sulphate: (**a**) after curing for 7 days, (**b**) after curing for 28 days, (**c**) after curing for 90 days.

**Figure 10 materials-14-05198-f010:**
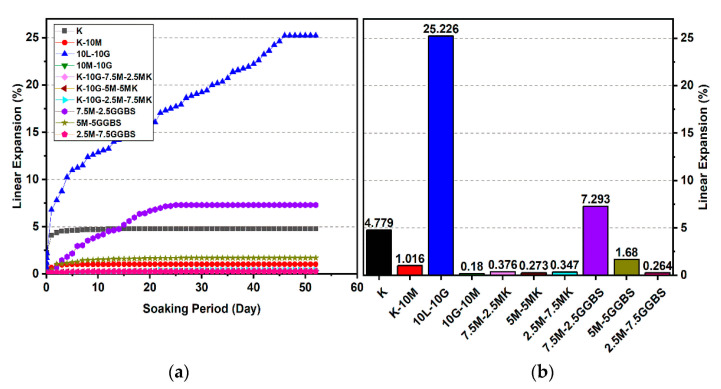
Vertical swelling strain of soils stabilized with 10% of L, M, M-MK, and M-GGBS after 7 days of curing: (**a**) results presented as a line curve, (**b**) results presented as columns.

**Figure 11 materials-14-05198-f011:**
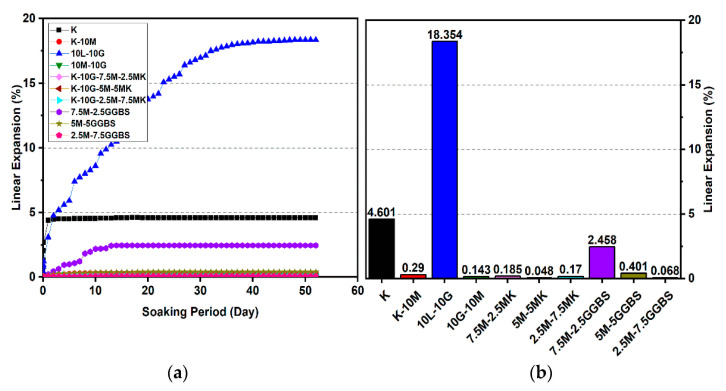
Vertical swelling strain of soils stabilized with 10% of L, M, M-MK, and M-GGBS after 28 days of curing: (**a**) results presented as a line curve, (**b**) results presented as columns.

**Figure 12 materials-14-05198-f012:**
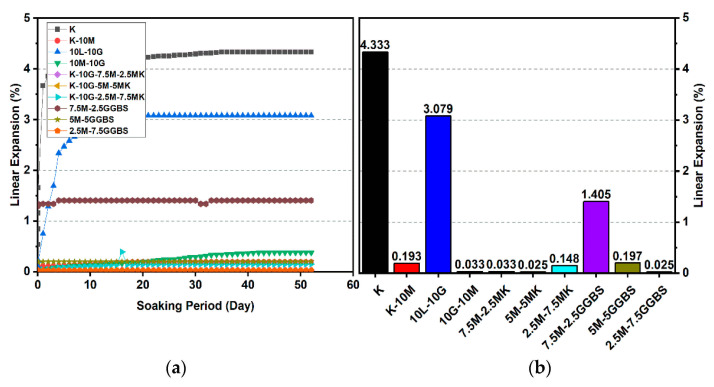
Vertical swelling strain of soils stabilized with 10% of L, M, M-MK, and M-GGBS after 90 days of curing: (**a**) results presented as a line curve, (**b**) results presented as columns.

**Figure 13 materials-14-05198-f013:**
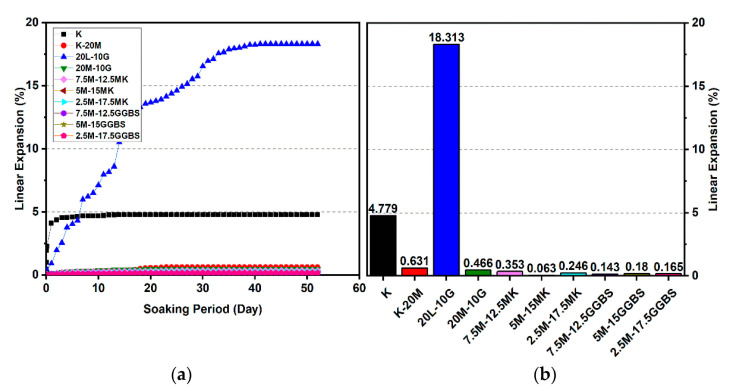
Vertical swelling strain of soils stabilized with 20% of L, M, M-MK, and M-GGBS after 7 days of curing: (**a**) results presented as a line curve, (**b**) results presented as columns.

**Figure 14 materials-14-05198-f014:**
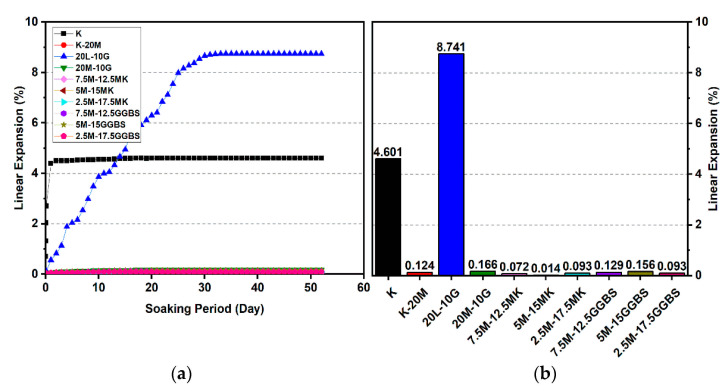
Vertical swelling strain of soils stabilized with 20% of L, M, M-MK, and M-GGBS after 28 days of curing: (**a**) results presented as a line curve, (**b**) results presented as columns.

**Figure 15 materials-14-05198-f015:**
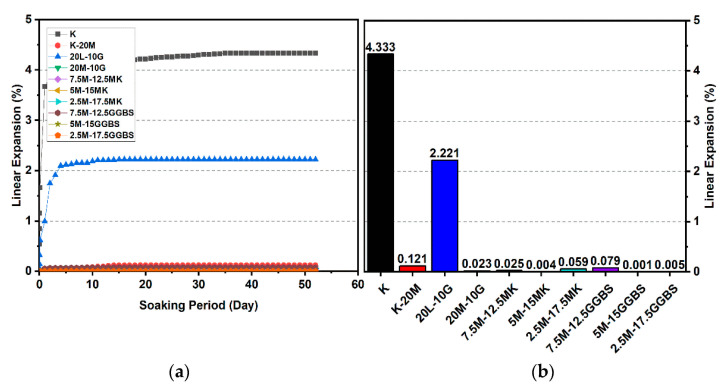
Vertical swelling strain of soils stabilized with 20% of L, M, M-MK, and M-GGBS after 90 days of curing: (**a**) results presented as a line curve, (**b**) results presented as columns.

**Figure 16 materials-14-05198-f016:**
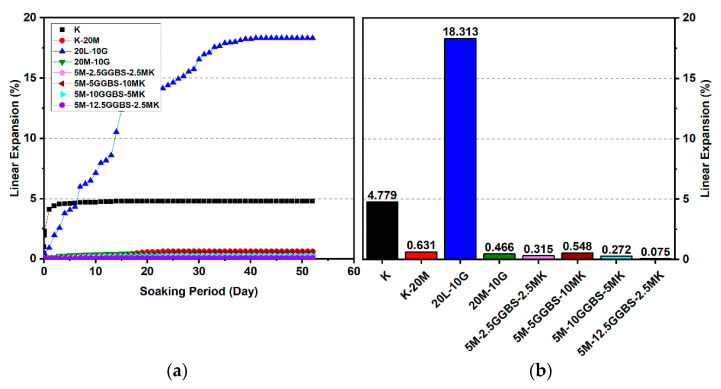
Vertical swelling strain of soils stabilized with 10% and 20% of L, M, and M-GGBS-MK after 7 days of curing: (**a**) results presented as a line curve, (**b**) results presented as columns.

**Figure 17 materials-14-05198-f017:**
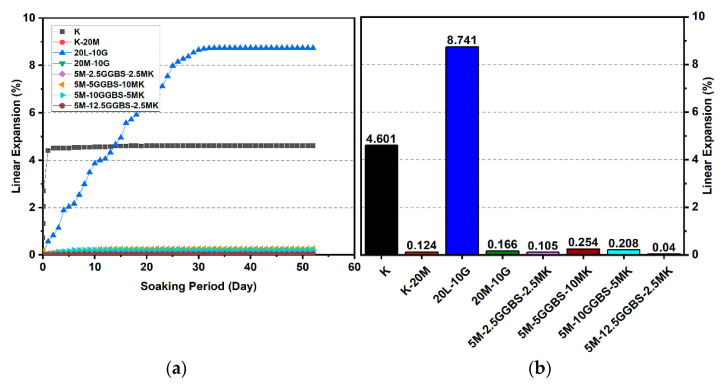
Vertical swelling strain of soils stabilized with 10% and 20% of L, M, and M-GGBS-MK after 28 days of curing: (**a**) results presented as a line curve, (**b**) results presented as columns.

**Figure 18 materials-14-05198-f018:**
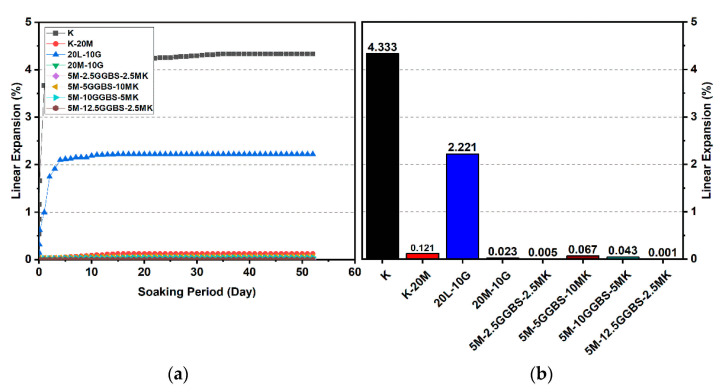
Vertical swelling strain of soils stabilized with 10% and 20% of L, M, and M-GGBS-MK after 90 days of curing: (**a**) results presented as a line curve, (**b**) results presented as columns.

**Figure 19 materials-14-05198-f019:**
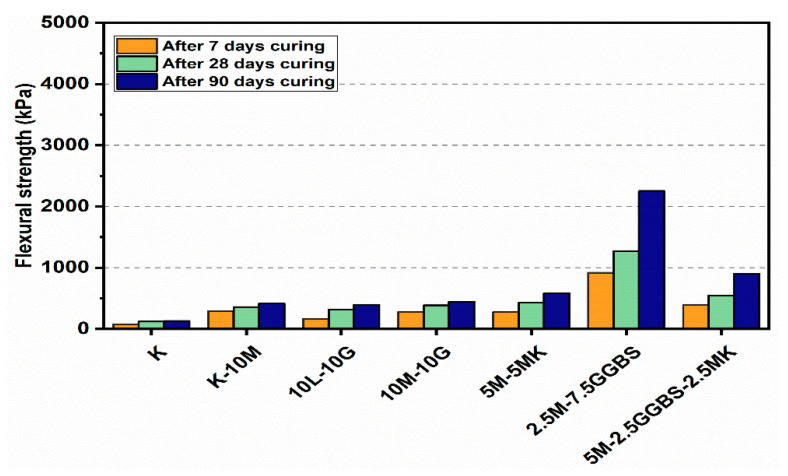
Flexural load curves of test gypseous soils treated with 10% mixture content after 7, 28 and 90 days of curing.

**Figure 20 materials-14-05198-f020:**
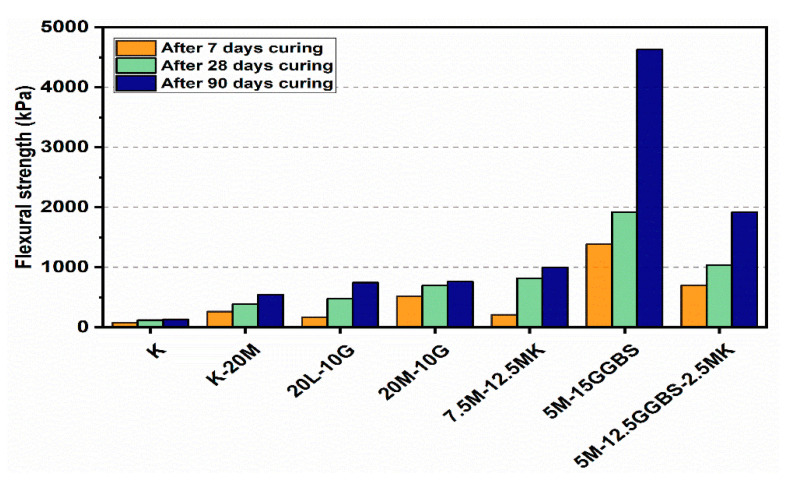
Flexural load curves of test gypseous soils treated with 20% mixture content after 7, 28 and 90 days of curing.

**Figure 21 materials-14-05198-f021:**
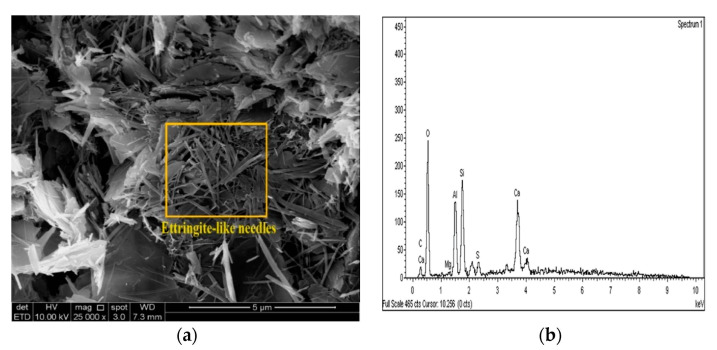
SEM photo and EDX spectrograph of stabilized gypseous soils after 90 days of curing: (**a**) SEM photo of 10L:10G and (**b**) EDX spectrograph of 10L:10G.

**Figure 22 materials-14-05198-f022:**
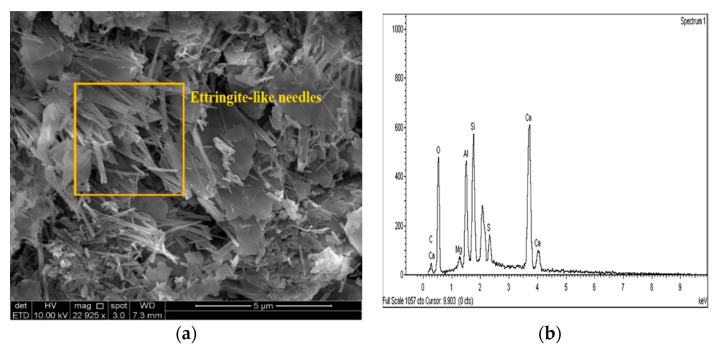
SEM photo and EDX spectrograph of stabilized gypseous soils after 90 days of curing: (**a**) SEM photo of 20L:10G and (**b**) EDX spectrograph of 20L:10G.

**Figure 23 materials-14-05198-f023:**
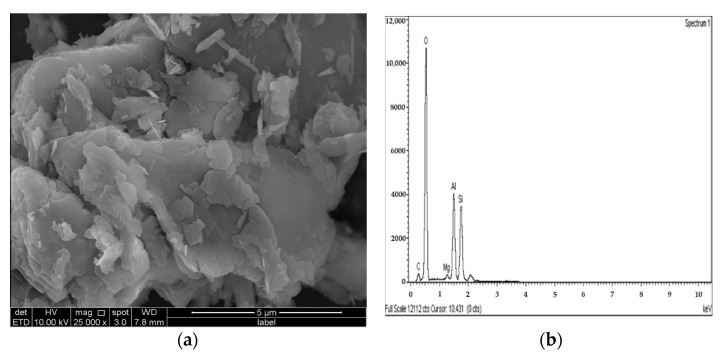
SEM photo and EDX spectrograph of stabilized gypseous soils after 90 days of curing: (**a**) SEM photo of 10M:10G and (**b**) EDX spectrograph of 10M:10G.

**Figure 24 materials-14-05198-f024:**
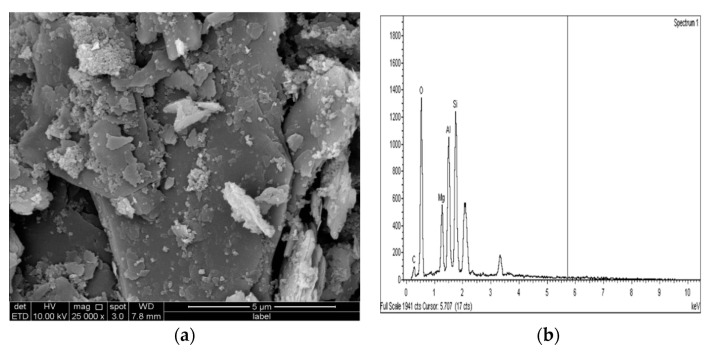
SEM photo and EDX spectrograph of stabilized gypseous soils after 90 days of curing: (**a**) SEM photo of 20M:10G and (**b**) EDX spectrograph of 20M:10G.

**Figure 25 materials-14-05198-f025:**
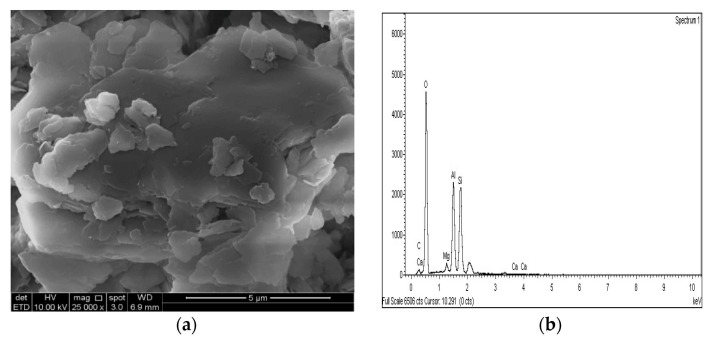
SEM photo and EDX spectrograph of stabilized gypseous soils after 90 days of curing: (**a**) SEM photo of 7.5M:12.5MK and (**b**) EDX spectrograph of 7.5M:12.5MK.

**Figure 26 materials-14-05198-f026:**
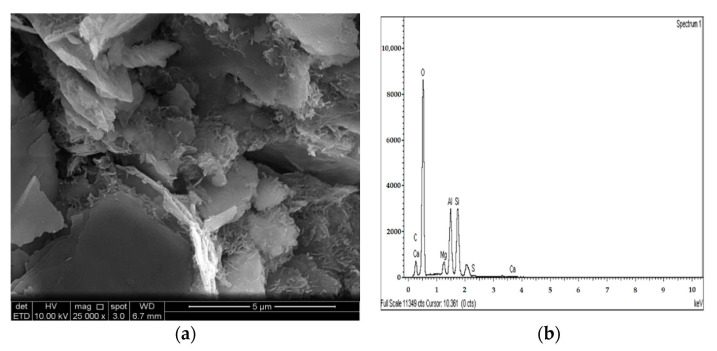
SEM photo and EDX spectrograph of stabilized gypseous soils after 90 days of curing: (**a**) SEM photo of 5M:15GGBS and (**b**) EDX spectrograph of 5M:15GGBS.

**Figure 27 materials-14-05198-f027:**
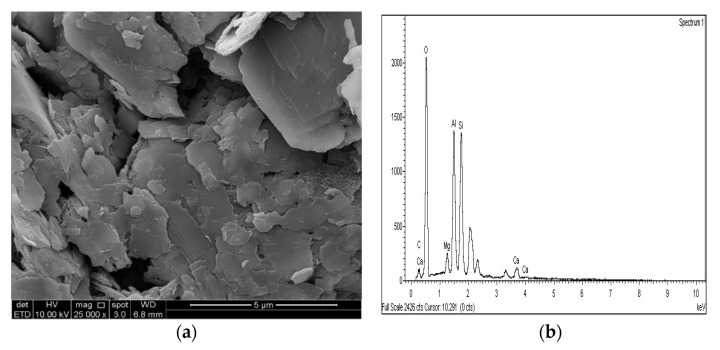
SEM photo and EDX spectrograph of stabilized gypseous soils after 90 days of curing: (**a**) SEM photo of 5M:12.5GGBS:2.5MK and (**b**) EDX spectrograph of 5M-12.5GGBS:2.5MK.

**Table 1 materials-14-05198-t001:** Characteristics of the kaolin clay used in this study.

Properties	Value
Liquid limit (%)	57.78
Plastic limit (%)	38.13
Shrinking limit (%)	4.4
Plasticity index	19.65
Sand (%)	-
Silt (%)	88.35
Clay (%)	11.65
Electric conductivity (µS/cm)	320
pH	5
Specific gravity (G_s_)	2.46
Water content (%)	1.01
Optimum moisture content (%)	29
Maximum dry density (Mg/m^3^)	1.326

**Table 2 materials-14-05198-t002:** Most common types of sulphates found in soil.

Sulphate Type	Common Name	Chemical Formula	Solubility
Calcium	Selenite, gypsum	CaSO_4_∙2H_2_O	1.44
Potassium	Arcanite	K_2_SO_4_	130
Magnesium	Epsomite	MgSO_4_∙7H_2_O	225
Sodium	Themadite, mirabilite	Na_2_SO_4_∙10H_2_O	>225

**Table 3 materials-14-05198-t003:** Chemical compositions of kaolin clay, L, M, G, GGBS, and MK.

Oxides	Characteristic (%)					
	Kaolin Clay	L	M	G	GGBS	MK
CaO	-	-	-		37	0.2
CaOH_2_	-	92	-		-	-
SiO_2_	58	2.5	-		32.7	52
Al_2_O_3_	38	0.9	-		15.3	36
Ca_2_SO_4_	-	0.1	-	99	-	-
SO_3_	-	-	0.03		4.7	-
MgO	-	3.5	99.5		8.1	0.1
Cl	-	-	0.01	0.005	-	-
Fe	-	0.06	0.01	0.005	-	8
H_2_O	-	0.7	0.2		-	-
Loss on ignition	11–14	0.24	0.25	0.99	2.2	3.7
pH	5	11.85	10.83	7.5	10.23	6.71
Specific gravity	2.46	2.23	3.58	2.34	2.96	2.33

**Table 4 materials-14-05198-t004:** Severity of sulphate levels.

Risk Level	Sulphate Concentration	
	Parts per Million	Percentage of Dry Weight
Low risk	>3000 ppm	>3%
Moderate risk	3000–5000 ppm	3–5%
Moderate to high risk	5000–8000 ppm	5–8%
High to unacceptable risk	>8000 ppm	>8%
Unacceptable risk	>10,000 ppm	>10%

**Table 5 materials-14-05198-t005:** Results of compaction test.

Stabilizer			MDD (Mg/m^3^)		OMC (%)	
Type	Binder Ratio (%)		Dosage (%)		Dosage (%)	
	10%	20%	10%	20%	10%	20%
K	0	0	1.326	1.326	29	29
**Unary**						
L	100	100	1.28	1.27	30	32
M	100	100	1.37	1.39	29	29.5
**Binary**						
M-MK	7.5:2.5	2.5:17.2	1.335	1.32	28	29.5
	5:5	5:15	1.33	1.31	28.5	30
	2.5:7.5	7.5:12.5	1.315	1.3	29.5	30.4
M-GGBS	7.5:2.5	2.5:17.2	1.375	1.38	27.2	26
	5:5	5:15	1.38	1.39	25	24.5
	2.5:7.5	7.5:12.5	1.39	1.4	24	23.5
**Ternary**						
M-GGBS-MK	5:2.5:2.5	5:5:10	1.35	1.36	25	26
		5:10:5	-	1.37	-	25.5
		5:12.5:2.5	-	1.38	-	25

**Table 6 materials-14-05198-t006:** Mixture designs of stabilizer agents.

Mix Code	Binder Composition	Binder Ratio (%)	Dosage (%)
Unary			
K	K	-	-
K-L	L	100	10, 20
K-M	M	100	10, 20
K-L-G	L	100	10, 20
K-M-G	M	100	10, 20
Binary			
K-M-MK-G	M:MK	3:1, 1:1, 1:3	10, 20
K-M-GGBS-G	M:GGBS	3:1, 1:1, 1:3	10, 20
Ternary			
K-M-GGBS-MK-G	M:GGBS:MK	1:0.5:0.5, 1:1:2, 1:2:1, 1:2.5:0.5	10, 20

(K) Kaolin, (G) Gypsum, (L) hydrated lime, (M) Nano-magnesium oxide, (MK) Metakaolin, (GGBS) Ground granulated blast-furnace slag.

**Table 7 materials-14-05198-t007:** Levels of swelling for clay soil.

Swelling (%)	Swelling Level
0	No swell
0–0.1	Negligible
0.1–0.5	Light
0.5–1.0	Medium
1.0–2.0	Strong
Over 2.0	Very strong

## Data Availability

Not applicable.

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
