# Peer review of "Novel Approach to the Treatment of Gypseous Soil-Induced Ettringite Using Blends of Non-Calcium-Based Stabilizer, Ground Granulated Blast-Furnace Slag, and Metakaolin"

_materials, 2021, doi:10.3390/ma14185198_

Round 1

Reviewer 1 Report

The authors proposed a new method to stabilize soils subjected to internal sulphate attack by the gypsum (CaSO4) by using combined of nano-MgO, MK (metakaolin), and GGBS (ground granulated blast-furnace slag). Different tests (flexural strength (FS), linear expansion (LP) tests, and unconfined compressive strength (UCS)) were performed before and after exposure to water. Effective agents for the stabilization of gypseous soil were clarified based on the test results. It seems that the proposed method is useful for improving the gypseous soil. Some modifications will be required.

Reviewer 2 Report

The paper reports an interesting and very useful experimental work. The manuscript is well structured and can be published after some revisions. The reviewer enjoyed reading this paper.

The manuscript has some weaknesses. Mentioned below aspects should be taken into consideration during the revision:

  1. Figures:

- Figure 2. The plot is of poor quality. I suggest using a lower density of the auxiliary grid lines; no fill in markers; other treatments to increase plot transparency.

- Figure 10 to 18. I suggest you reduce the range of the Y axis to use the entire plot area; The legend contains more items than the displayed waveforms (as seen in the previous note); other treatments to increase plot transparency.

  1. Units and abbreviations:

I suggest adding "Nomenclature" (as list of symbols, list of abbreviations and subscripts and others) in the manuscript. Authors use a lot of abbreviations in the content, so I suggest using the nomenclature.
